# Linguistically Motivated Sign Language Segmentation

**Amit Moryossef**[*1,2], **Zifan Jiang**[*2]
**Mathias Müller**[2], **Sarah Ebling**[2], **Yoav Goldberg**[1]
[1]Bar-Ilan University, [2]University of Zurich
amitmoryossef@gmail.com, jiang@cl.uzh.ch

## Abstract

Sign language segmentation is a crucial task in sign language processing systems. It enables downstream tasks such as sign recognition, transcription, and machine translation. In this work, we consider two kinds of segmentation: segmentation into individual signs and segmentation into *phrases*, larger units comprising several signs. We propose a novel approach to jointly model these two tasks.

Our method is motivated by linguistic cues observed in sign language corpora. We replace the predominant IO tagging scheme with BIO tagging to account for continuous signing. Given that prosody plays a significant role in phrase boundaries, we explore the use of optical flow features. We also provide an extensive analysis of hand shapes and 3D hand normalization.

We find that introducing BIO tagging is necessary to model sign boundaries. Explicitly encoding prosody by optical flow improves segmentation in shallow models, but its contribution is negligible in deeper models. Careful tuning of the decoding algorithm atop the models further improves the segmentation quality.

We demonstrate that our final models generalize to out-of-domain video content in a different signed language, even under a zero-shot setting. We observe that including optical flow and 3D hand normalization enhances the robustness of the model in this context.

## 1 Introduction

Signed languages are natural languages used by deaf and hard-of-hearing individuals to communicate through a combination of manual and non-manual elements ([Sandler and Lillo-Martin, 2006](#)). Like spoken languages, signed languages have their own distinctive grammar, and vocabulary, that have evolved through natural processes of language development ([Sandler, 2010](#)).

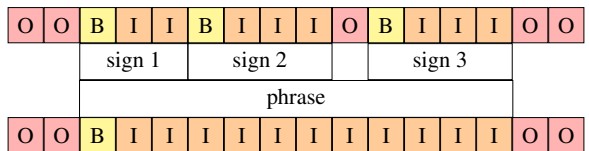

Figure 1: Per-frame classification of a sign language utterance following a BIO tagging scheme. Each box represents a single frame of a video. We propose a joint model to segment *signs* (top) and *phrases* (bottom) at the same time. B=beginning, I=inside, O=outside. The figure illustrates continuous signing where signs often follow each other without an O frame between them.

Sign language transcription and translation systems rely on the accurate temporal segmentation of sign language videos into meaningful units such as signs ([Santemiz et al., 2009](#); [Renz et al., 2021a](#)) or signing sequences corresponding to subtitle units[1] ([Bull et al., 2020b](#)). However, sign language segmentation remains a challenging task due to the difficulties in defining meaningful units in signed languages ([De Sisto et al., 2021](#)). Our approach is the first to consider two kinds of units in one model. We simultaneously segment single signs and phrases (larger units) in a unified framework.

Previous work typically approached segmentation as a binary classification task (including segmentation tasks in audio signal processing and computer vision), where each frame/pixel is predicted to be either part of a segment or not. However, this approach neglects the intricate nuances of continuous signing, where segment boundaries are not strictly binary and often blend in reality. One sign or phrase can immediately follow another, transitioning smoothly, without a frame between them being distinctly *outside* (Figure 1 and §3.1).

We propose incorporating linguistically motivated cues to address these challenges and improve sign language segmentation. To cope with contin-

---

[1]Subtitles may not always correspond directly to sentences. They frequently split within a sentence and could be temporally offset from the corresponding signing segments.

---

[*]Equal contribution authors.

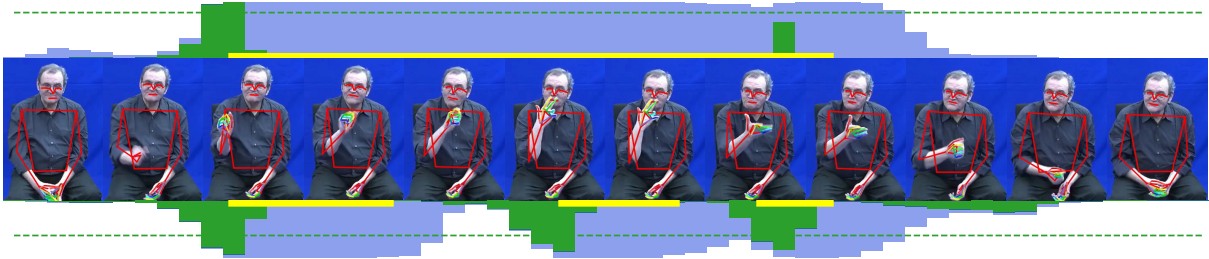

Figure 2: The annotation of the first phrase in a video from the test set (dgskorpus_goe_02), in yellow, signing: "Why do you smoke?" through the use of three signs: *WHY* (+mouthed), *TO-SMOKE*, and a gesture (+mouthed) towards the other signer. At the top, our phrase segmentation model predicts a single phrase that initiates with a B tag (in green) above the B-threshold (green dashed line), followed by an I (in light blue), and continues until falling below a certain threshold. At the bottom, our sign segmentation model accurately segments the three signs.

uous signing, we adopt a BIO-tagging approach (Ramshaw and Marcus, 1995), where in addition to predicting a frame to be *in* or *out* of a segment, we also classify the *beginning* of the segment as shown in Figure 2. Since phrase segmentation is primarily marked with prosodic cues (i.e., pauses, extended sign duration, facial expressions) (Sandler, 2010; Ormel and Crasborn, 2012), we explore using optical flow to explicitly model them (§3.2). Since signs employ a limited number of hand shapes, we additionally perform 3D hand normalization (§3.3).

Evaluating on the Public DGS Corpus (Prillwitz et al., 2008; Hanke et al., 2020) (DGS stands for German Sign Language), we report enhancements in model performance following specific modifications. We compare our final models after hyperparameter optimization, including parameters for the decoding algorithm, and find that our best architecture using only the poses is comparable to the one that uses optical flow and hand normalization.

Reassuringly, we find that our model generalizes when evaluated on additional data from different signed languages in a zero-shot approach. We obtain segmentation scores that are competitive with previous work and observe that incorporating optical flow and hand normalization makes the model more robust for out-of-domain data.

Lastly, we conduct an extensive analysis of pose-based hand manipulations for signed languages (Appendix C). Despite not improving our segmentation model due to noise from current 3D pose estimation models, we emphasize its potential value for future work involving skeletal hand poses. Based on this analysis, we propose several measurable directions for improving 3D pose estimation.

Our code and models are available at `https://github.com/sign-language-processing/transcription`.

## 2 Related Work

### 2.1 Sign Language Detection

Sign language detection (Borg and Camilleri, 2019; Moryossef et al., 2020; Pal et al., 2023) is the task of determining whether signing activity is present in a given video frame. A similar task in spoken languages is voice activity detection (VAD) (Sohn et al., 1999; Ramırez et al., 2004), the detection of when human voice is used in an audio signal. As VAD methods often rely on speech-specific representations such as spectrograms, they are not necessarily applicable to videos.

Borg and Camilleri (2019) introduced the classification of frames taken from YouTube videos as either signing or not signing. They took a spatial and temporal approach based on VGG-16 (Simonyan and Zisserman, 2015) CNN to encode each frame and used a Gated Recurrent Unit (GRU) (Cho et al., 2014) to encode the sequence of frames in a window of 20 frames at 5fps. In addition to the raw frame, they either encoded optical-flow history, aggregated motion history, or frame difference.

Moryossef et al. (2020) improved upon their method by performing sign language detection in real time. They identified that sign language use involves movement of the body and, as such, designed a model that works on top of estimated human poses rather than directly on the video signal. They calculated the optical flow norm of every joint detected on the body and applied a shallow yet effective contextualized model to predict for every frame whether the person is signing or not.

While these recent detection models achieve high performance, we need well-annotated data including interference and non-signing distractions for proper real-world evaluation. Pal et al. (2023) conducted a detailed analysis of the impact of

signer overlap between the training and test sets on two sign detection benchmark datasets (Signing in the Wild (Borg and Camilleri, 2019) and the DGS Corpus (Hanke et al., 2020)) used by Borg and Camilleri (2019) and Moryossef et al. (2020). By comparing the accuracy with and without overlap, they noticed a relative decrease in performance for signers not present during training. As a result, they suggested new dataset partitions that eliminate overlap between train and test sets and facilitate a more accurate evaluation of performance.

## 2.2 Sign Language Segmentation

Segmentation consists of detecting the frame boundaries for signs or phrases in videos to divide them into meaningful units. While the most canonical way of dividing a spoken language text is into a linear sequence of words, due to the simultaneity of sign language, the notion of a sign language "word" is ill-defined, and sign language cannot be fully linearly modeled.

Current methods resort to segmenting units loosely mapped to signed language units (Santemiz et al., 2009; Farag and Brock, 2019; Bull et al., 2020b; Renz et al., 2021a,b; Bull et al., 2021) and do not explicitly leverage reliable linguistic predictors of sentence boundaries such as prosody in signed languages (i.e., pauses, extended sign duration, facial expressions) (Sandler, 2010; Ormel and Crasborn, 2012). De Sisto et al. (2021) call for a better understanding of sign language structure, which they believe is the necessary ground for the design and development of sign language recognition and segmentation methodologies.

Santemiz et al. (2009) automatically extracted isolated signs from continuous signing by aligning the sequences obtained via speech recognition, modeled by Dynamic Time Warping (DTW) and Hidden Markov Models (HMMs) approaches.

Farag and Brock (2019) used a random forest classifier to distinguish frames containing signs in Japanese Sign Language based on the composition of spatio-temporal angular and distance features between domain-specific pairs of joint segments.

Bull et al. (2020b) segmented French Sign Language into segments corresponding to subtitle units by relying on the alignment between subtitles and sign language videos, leveraging a spatio-temporal graph convolutional network (STGCN; Yu et al. (2017)) with a BiLSTM on 2D skeleton data.

Renz et al. (2021a) located temporal bound-aries between signs in continuous sign language videos by employing 3D convolutional neural network representations with iterative temporal segment refinement to resolve ambiguities between sign boundary cues. Renz et al. (2021b) further proposed the Changepoint-Modulated Pseudo-Labelling (CMPL) algorithm to solve the problem of source-free domain adaptation.

Bull et al. (2021) presented a Transformer-based approach to segment sign language videos and align them with subtitles simultaneously, encoding subtitles by BERT (Devlin et al., 2019) and videos by CNN video representations.

## 3 Motivating Observations

To motivate our proposed approach, we make a series of observations regarding the intrinsic nature of sign language expressions. Specifically, we highlight the unique challenges posed by the continuous flow of sign language expressions (§3.1), the role of prosody in determining phrase boundaries (§3.2), and the influence of hand shape changes in indicating sign boundaries (§3.3).

### 3.1 Boundary Modeling

When examining the nature of sign language expressions, we note that the utterances are typically signed in a continuous flow, with minimal to no pauses between individual signs. This continuity is particularly evident when dealing with lower frame rates. This continuous nature presents a significant difference from *text* where specific punctuation marks serve as indicators of phrase boundaries, and a semi-closed set of tokens represent the *words*.

Given these characteristics, directly applying conventional segmentation or sign language detection models—that is, utilizing IO tagging in a manner similar to image or audio segmentation models—may not yield the optimal solution, particularly at the sign level. Such models often fail to precisely identify the boundaries between signs.

A promising alternative is the Beginning-Inside-Outside (BIO) tagging (Ramshaw and Marcus, 1995). BIO tagging was originally used for named entity recognition, but its application extends to any sequence chunking task beyond the text modality. In the context of sign language, BIO tagging provides a more refined model for discerning boundaries between signs and phrases, thus significantly improving segmentation performance (Figure 1).

To test the viability of the BIO tagging approach

in comparison with the IO tagging, we conducted an experiment on the Public DGS Corpus. The entire corpus was transformed to various frame rates and the sign segments were converted to frames using either BIO or IO tagging, then decoded back into sign segments. Figure 4 illustrates the results of this comparison. Note that the IO tagging was unable to reproduce the same number of segments as the BIO tagging on the gold data. This underscores the importance of BIO tagging in identifying sign and phrase boundaries.

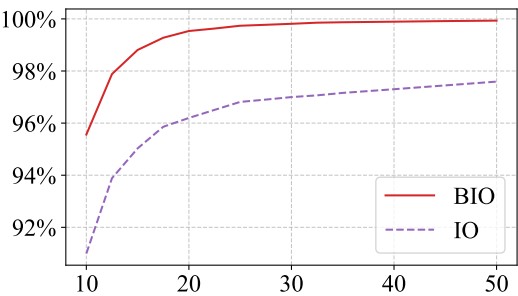

Figure 4: Reproduced sign segments in the Public DGS Corpus by BIO and IO tagging at various frame rates. 99.7% of segments reproduced at 25fps by BIO tagging.

## 3.2 Phrase Boundaries

Linguistic research has shown that prosody is a reliable predictor of phrase boundaries in signed languages (Sandler, 2010; Ormel and Crasborn, 2012). We observe that this is also the case in the Public DGS Corpus dataset used in our experiments. To illustrate this, we model pauses and movement using optical flow directly on the poses as proposed by Moryossef et al. (2020). Figure 3 demonstrates that a change in motion signifies the presence of a pause, which, in turn, indicates a phrase boundary.

## 3.3 Sign Boundaries

We observe that signs generally utilize a limited number of hand shapes, with the majority of signs utilizing a maximum of two hand shapes. For example, linguistically annotated datasets, such as ASL-LEX (Sehyr et al., 2021) and ASLLVD (Neidle et al., 2012), only record one initial hand shape per hand and one final hand shape. Mandel (1981, p. 87) argued that there can only be one set of selected fingers per sign, constraining the number of handshapes in signs. This limitation is referred to as the *Selected Fingers Constraint*. And indeed, Sandler et al. (2008) find that the optimal form of a sign is monosyllabic, and that handshape change is organized by the syllable unit.

To illustrate this constraint empirically, we show a histogram of hand shapes per sign in SignBank[2] for 705, 151 signs in Figure 5.

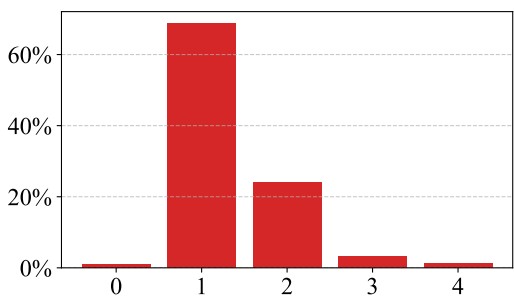

Figure 5: Number of hand shapes per sign in SignBank.

Additionally, we found that a change in the dominant hand shape often signals the presence of a sign boundary. Specifically, out of 27, 658 sentences, comprising 354, 955 pairs of consecutive signs, only 17.38% of consecutive signs share the same base hand shape[3]. Based on these observations, we propose using 3D hand normalization as an indicative cue for hand shapes to assist in detecting sign boundaries. We hypothesize that performing 3D hand normalization makes it easier for

---

[2]https://signbank.org/signpuddle2.0/

[3]It is important to note that this percentage is inflated, as it may encompass overlaps across the dominant and non-dominant hands, which were not separated for this analysis.

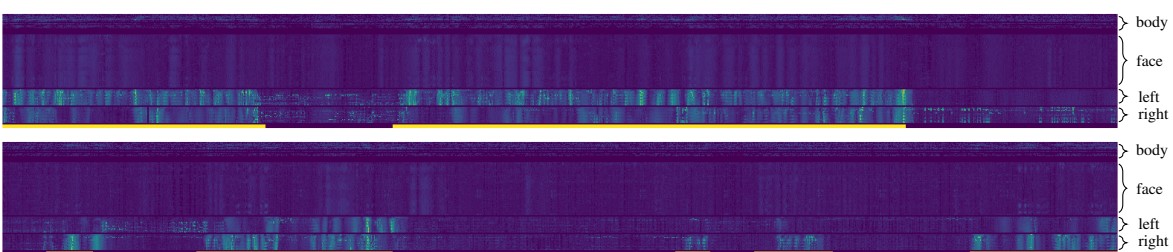

Figure 3: Optical flow for a conversation between two signers (signer 1 top, signer 2 bottom). The x-axis is the progression across 30 seconds. The yellow marks the annotated phrase spans. (Source: Moryossef et al. (2020))

the model to extract the hand shape. We expand on this process and show examples in Appendix C.

## 4 Experimental Setup

In this section, we describe the experimental setup used to evaluate our linguistically motivated approach for sign language segmentation. This includes a description of the Public DGS Corpus dataset used in the study, the methodology employed to perform sign and phrase segmentation, and the evaluation metrics used to measure the performance of the proposed approach.

### 4.1 Dataset

The Public DGS Corpus (Prillwitz et al., 2008; Hanke et al., 2020) is a distinctive sign language dataset that includes both accurate sign-level annotation from continuous signing, and well-aligned phrase-level translation in spoken language.

The corpus comprises 404 documents / 714 videos[4] with an average duration of 7.55 minutes, featuring either one signer or two signers, at 50 fps. Most of these videos feature gloss transcriptions and spoken language translations (German and English), except for the ones in the "Joke" category, which are not annotated and thus excluded from our model[5]. The translations are comprised of full spoken language paragraphs, sentences, or phrases (i.e., independent/main clauses).

Each gloss span is considered a gold sign segment, following a tight annotation scheme (Hanke et al., 2012). Phrase segments are identified by examining every translation, with the segment assumed to span from the start of its first sign to the end of its last sign, correcting imprecise annotation.

This corpus is enriched with full-body pose estimations from OpenPose (Cao et al., 2019; Schulder and Hanke, 2019) and Mediapipe Holistic (Grishchenko and Bazarevsky, 2020). We use the *3.0.0-uzh-document* split from Zhang et al. (2023). After filtering the unannotated data, we are left with 296 documents / 583 videos for training, 6 / 12 for validation, and 9 / 17 for testing. The mean number of signs and phrases in a video from the training set is 613 and 111, respectively.

---

[4]The number of videos is nearly double the number of documents because each document typically includes two signers, each of whom produces one video for segmentation.

[5]We also exclude documents with missing annotations. $id \in \{1289910, 1245887, 1289868, 1246064, 1584617\}$

### 4.2 Methodology

Our proposed approach for sign language segmentation is based on the following steps:

1. **Pose Estimation** Given a video, we first adjust it to 25 fps and estimate body poses using the MediaPipe Holistic pose estimation system. We do not use OpenPose because it lacks a $Z$-axis, which prevents 3D rotation used for hand normalization. The shape of a pose is represented as (frames $\times$ keypoints $\times$ axes).

2. **Pose Normalization** To generalize over video resolution and distance from the camera, we normalize each of these poses such that the mean distance between the shoulders of each person equals 1, and the mid-point is at $(0, 0)$ (Celebi et al., 2013). We also remove the legs since they are less relevant to signing.

3. **Optical Flow** We follow the equation in Moryossef et al. (2020, Equation 1).

4. **3D Hand Normalization** We rotate and scale each hand to ensure that the same hand shape is represented in a consistent manner across different frames. We rotate the 21 $XYZ$ keypoints of the hand so that the back of the hand lies on the $XY$ plane, we then rotate the hand so that the metacarpal bone of the middle finger lies on the $Y$-axis, and finally, we scale the hand such that the bone is of constant length. Visualizations are presented in Appendix C.

5. **Sequence Encoder** For every frame, the pose is first flattened and projected into a standard dimension (256), then fed through an LSTM encoder (Hochreiter and Schmidhuber, 1997).

6. **BIO Tagging** On top of the encoder, we place two BIO classification heads for sign and phrase independently. *B* denotes the beginning of a sign or phrase, *I* denotes the middle of a sign or phrase, and *O* denotes being outside a sign or phrase. Our cross-entropy loss is proportionally weighted in favor of *B* as it is a *rare* label[6] compared to *I* and *O*.

7. **Greedy Segment Decoding** To decode the frame-level BIO predictions into sign/phrase segments, we define a segment to start with the first frame possessing a *B* probability

---

[6]B:I:O is about 1:5:18 for signs and 1:58:77 for phrases.

greater than a predetermined threshold (defaulted at 0.5). The segment concludes with the first frame among the subsequent frames, having either a *B* or *O* probability exceeding the threshold. We provide the pseudocode of the decoding algorithm in Appendix B.

## 4.3 Experiments

Our approach is evaluated through a series of six sets of experiments. Each set is repeated three times with varying random seeds. Preliminary experiments were conducted to inform the selection of hyperparameters and features, the details of which can be found in Table 3 in Appendix A. Model selection is based on validation metrics.

1. **E0: IO Tagger** We re-implemented and reproduced[7] the sign language detection model proposed by Moryossef et al. (2020), in PyTorch (Paszke et al., 2019) as a naive baseline. This model processes optical flow as input and outputs *I* (is signing) and *O* (not signing) tags.

2. **E1: Bidirectional BIO Tagger** We replace the IO tagging heads in *E0* with BIO heads to form our baseline. Our preliminary experiments indicate that inputting only the 75 hand and body keypoints and making the LSTM layer bidirectional yields optimal results.

3. **E2: Adding Reduced Face Keypoints** Although the 75 hand and body keypoints serve as an efficient minimal set for sign language detection/segmentation models, we investigate the impact of other nonmanual sign language articulators, namely, the face. We introduce a reduced set of 128 face keypoints that signify the signer's *face contour*[8].

4. **E3: Adding Optical Flow** At every time step $t$ we append the optical flow between $t$ and $t-1$ to the current pose frame as an additional dimension after the $XYZ$ axes.

5. **E4: Adding 3D Hand Normalization** At every time step, we normalize the hand poses and concatenate them to the current pose.

6. **E5: Autoregressive Encoder** We replace the encoder with the one proposed by Jiang et al. (2023) for the detection and classification of great ape calls from raw audio signals. Specifically, we add autoregressive connections between time steps to encourage consistent output labels. The logits at time step $t$ are concatenated to the input of the next time step, $t+1$. This modification is implemented bidirectionally by stacking two autoregressive encoders and adding their output up before the Softmax operation. However, this approach is inherently slow, as we have to fully wait for the previous time step predictions before we can feed them to the next time step.

## 4.4 Evaluation Metrics

We evaluate the performance of our proposed approach for sign and phrase segmentation using the following metrics:

- **Frame-level F1 Score** For each frame, we apply the *argmax* operation to make a local prediction of the BIO class and calculate the macro-averaged per-class F1 score against the ground truth label. We use this frame-level metric during validation as the primary metric for model selection and early stopping, due to its independence from a potentially variable segment decoding algorithm (§5.2).

- **Intersection over Union (IoU)** We compute the IoU between the ground truth segments and the predicted segments to measure the degree of overlap. Note that we do not perform a one-to-one mapping between the two using techniques like DTW. Instead, we calculate the total IoU based on all segments in a video.

- **Percentage of Segments (%)** To complement IoU, we introduce the percentage of segments to compare the number of segments predicted by the model with the ground truth annotations. It is computed as follows: $\frac{\#\text{predicted segments}}{\#\text{ground truth segments}}$. The optimal value is 1.

- **Efficiency** We measure the efficiency of each model by the number of parameters and the training time of the model on a Tesla V100-SXM2-32GB GPU for 100 epochs[9].

---

[7]The initial implementation uses OpenPose (Cao et al., 2019), at 50 fps. Preliminary experiments reveal that these differences do not significantly influence the results.

[8]We reduce the dense *FACE_LANDMARKS* in Mediapipe Holistic to the contour keypoints according to the variable *mediapipe.solutions.holistic.FACEMESH_CONTOURS*.

[9]Exceptionally the autoregressive models in *E5* were trained on an NVIDIA A100-SXM4-80GB GPUA100 which doubles the training speed of V100, still the training is slow.

| Experiment | | Sign | | | Phrase | | | Efficiency | |
|---|---|---|---|---|---|---|---|---|---|
| | | **F1** | **IoU** | **%** | **F1** | **IoU** | **%** | **#Params** | **Time** |
| **E0** | **Moryossef et al. (2020)** | — | 0.46 | 1.09 | — | 0.70 | **1.00** | **102K** | **0:50:17** |
| **E1** | **Baseline** | 0.56 | 0.66 | 0.91 | 0.59 | 0.80 | 2.50 | 454K | 1:01:50 |
| **E2** | **E1 + Face** | 0.53 | 0.58 | 0.64 | 0.57 | 0.76 | 1.87 | 552K | 1:50:31 |
| **E3** | **E1 + Optical Flow** | 0.58 | 0.62 | 1.12 | 0.60 | 0.82 | 3.19 | 473K | 1:20:17 |
| **E4** | **E3 + Hand Norm** | 0.56 | 0.61 | 1.07 | 0.60 | 0.80 | 3.24 | 516K | 1:30:59 |
| **E1s** | **E1 + Depth=4** | **0.63** | **0.69** | 1.11 | **0.65** | 0.82 | 1.63 | 1.6M | 4:08:48 |
| **E2s** | **E2 + Depth=4** | 0.62 | **0.69** | 1.07 | 0.63 | 0.84 | 2.68 | 1.7M | 3:14:03 |
| **E3s** | **E3 + Depth=4** | 0.60 | 0.63 | 1.13 | 0.64 | 0.80 | 1.53 | 1.7M | 4:08:30 |
| **E4s** | **E4 + Depth=4** | 0.59 | 0.63 | 1.13 | 0.62 | 0.79 | 1.43 | 1.7M | 4:35:29 |
| **E1s\*** | **E1s + Tuned Decoding** | — | **0.69** | 1.03 | — | **0.85** | 1.02 | — | — |
| **E4s\*** | **E4s + Tuned Decoding** | — | 0.63 | 1.06 | — | 0.79 | 1.12 | — | — |
| **E5** | **E4s + Autoregressive** | 0.45 | 0.47 | 0.88 | 0.52 | 0.63 | 2.72 | 1.3M | ~3 days |

Table 1: Mean test evaluation metrics for our experiments. The best score of each column is in bold and a star (*) denotes further optimization of the decoding algorithm without changing the model (only affects *IoU* and *%*). Table 4 in Appendix A contains a complete report including validation metrics and standard deviation of all experiments.

## 5 Results and Discussion

We report the mean test evaluation metrics for our experiments in Table 1. We do not report F1 Score for *E0* since it has a different number of classes and is thus incomparable. Comparing *E1* to *E0*, we note that the model's bidirectionality, the use of poses, and BIO tagging indeed help outperform the model from previous work where only optical flow and IO tagging are used. While *E1* predicts an excessive number of phrase segments, the IoUs for signs and phrases are both higher.

Adding face keypoints (*E2*) makes the model worse, while including optical flow (*E3*) improves the F1 scores. For phrase segmentation, including optical flow increases IoU, but over-segments phrases by more than 300%, which further exaggerates the issue in *E1*. Including hand normalization (*E4*) on top of *E3* slightly deteriorates the quality of both sign and phrase segmentation.

From the non-exhaustive hyperparameter search in the preliminary experiments (Table 3), we examined different hidden state sizes (64, 128, 256, 512, 1024) and a range of 1 to 8 LSTM layers, and conclude that a hidden size of 256 and 4 layers with $1e-3$ learning rate are optimal for *E1*, which lead to *E1s*. We repeat the setup of *E2*, *E3*, and *E4* with these refined hyper-parameters, and show that all of them outperform their counterparts, notably in that they ease the phrase over-segmentation problem.

In *E2s*, we reaffirmed that adding face keypoints does not yield beneficial results, so we exclude face in future experiments. Although the face is an essential component to understanding sign language expressions and does play some role in sign and phrase level segmentation, we believe that the 128 face contour points are too dense for the model to learn useful information compared to the 75 body points, and may instead confuse the model.

In addition, the benefits of explicitly including optical flow (*E3s*) fade away with the increased model depth and we speculate that now the model might be able to learn the optical flow features by itself. Surprisingly, while adding hand normalization (*E4s*) still slightly worsens the overall results, it has the best phrase percentage.

From *E4s* we proceeded with the training of *E5*, an autoregressive model. Unexpectedly, counter to our intuition and previous work, *E5* underachieves our baseline across all evaluation metrics[10].

### 5.1 Challenges with 3D Hand Normalization

While the use of 3D hand normalization is well-justified in §3.3, we believe it does not help the model due to poor depth estimation quality,

---

[10]*E5* should have more parameters than *E4s*, but because of an implementation bug, each LSTM layer has half the parameters. Based on the current results, we assume that autoregressive connections (even with more parameters) will not improve our models.

as further corroborated by recent research from De Coster et al. (2023). Therefore, we consider it a negative result, showing the deficiencies in the 3D pose estimation system. The evaluation metrics we propose in Appendix C could help identify better pose estimation models for this use case.

## 5.2 Tuning the Segment Decoding Algorithm

We selected *E1s* and *E4s* to further explore the segment decoding algorithm. As detailed in §4.2 and Appendix B, the decoding algorithm has two tunable parameters, $threshold_b$ and $threshold_o$. We conducted a grid search with these parameters, using values from 10 to 90 in increments of 10. We additionally experimented with a variation of the algorithm that conditions on the most likely class by *argmax* instead of fixed threshold values, which turned out similar to the default version.

We only measured the results using IoU and the percentage of segments at validation time since the F1 scores remain consistent in this case. For sign segmentation, we found using $threshold_b = 60$ and $threshold_o = 40/50/60$ yields slightly better results than the default setting (50 for both). For phrase segmentation, we identified that higher threshold values ($threshold_b = 90, threshold_o = 90$ for *E1s* and $threshold_b = 80, threshold_o = 80/90$ for *E4s*) improve on the default significantly, especially on the percentage metric. We report the test results under *E1s\** and *E4s\**, respectively.

Despite formulating a single model, we underline a separate sign/phrase model selection process to archive the best segmentation results. Figure 6 illustrates how higher threshold values reduce the number of predicted segments and skew the distribution of predicted phrase segments towards longer ones in *E1s/E1s\**. As Bull et al. (2020b) suggest, advanced priors could also be integrated into the decoding algorithm.

## 5.3 Comparison to Previous Work

We re-implemented and re-purposed the sign language detection model introduced in Moryossef et al. (2020) for our segmentation task as a baseline since their work is the state-of-the-art and the only comparable model designed for the Public DGS Corpus dataset. As a result, we show the necessity of replacing IO tagging with BIO tagging to tackle the subtle differences between the two tasks.

For *phrase* segmentation, we compare to Bull et al. (2020b). We note that our definition of sign language phrases (spanning from the start of its first

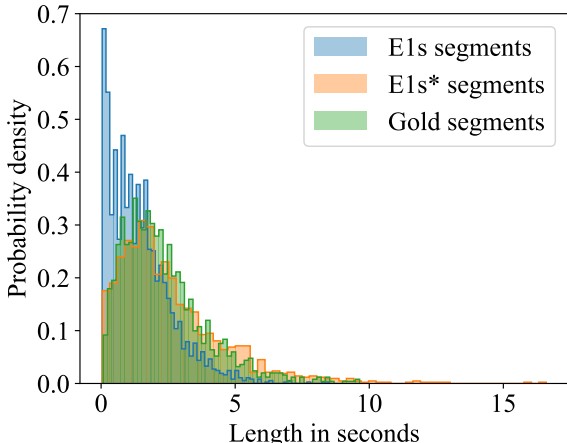

Figure 6: Probability density of phrase segment lengths.

sign to the end of its last sign) is tighter than the subtitle units used in their paper and that we use different training datasets of different languages and domains. Nevertheless, we implemented some of their frame-level metrics and show the results in Table 2 on both the Public DGS Corpus and the MEDIAPI-SKEL dataset (Bull et al., 2020a) in French Sign Language (LSF). We report both zero-shot out-of-domain results[11] and the results of our models trained specifically on their dataset without the spatio-temporal graph convolutional network (ST-GCN) (Yan et al., 2018) used in their work for pose encoding.

| Data | Model | ROC-AUC | F1-M |
|------|-------|---------|------|
| LSF | **full (theirs)** | 0.87 | — |
| | **body (theirs)** | 0.87 | — |
| | **E1s (ours, zero-shot)** | 0.71 | 0.41 |
| | **E4s (ours, zero-shot)** | 0.76 | 0.44 |
| | **E1s (ours, trained)** | 0.87 | 0.49 |
| | **E4s (ours, trained)** | 0.87 | 0.51 |
| DGS | **E1s (ours)** | 0.91 | 0.65 |
| | **E4s (ours)** | 0.90 | 0.62 |

Table 2: Evaluation metrics used in Bull et al. (2020b). *ROC-AUC* is applied exclusively on the *O*-tag. For comparison *F1-M* denotes the macro-averaged per-class F1 used in this work across all tags. The first two rows are the best results taken from Table 1 in their paper. The next four rows represent how our models perform on their data in a zero-shot setting, and in a supervised setting, and the last two rows represent how our models perform on our data.

---

[11]The zero-shot results are not directly comparable to theirs due to different datasets and labeling approaches.

For *sign* segmentation, we do not compare to Renz et al. (2021a,b) due to different datasets and the difficulty in reproducing their segment-level evaluation metrics. The latter depends on the decoding algorithm and a way to match the gold and predicted segments, both of which are variable.

# 6 Conclusions

This work focuses on the automatic segmentation of signed languages. We are the first to formulate the segmentation of individual signs and larger sign phrases as a joint problem.

We propose a series of improvements over previous work, linguistically motivated by careful analyses of sign language corpora. Recognizing that sign language utterances are typically continuous with minimal pauses, we opted for a BIO tagging scheme over IO tagging. Furthermore, leveraging the fact that phrase boundaries are marked by prosodic cues, we introduce optical flow features as a proxy for prosodic processes. Finally, since signs typically employ a limited number of hand shapes, to make it easier for the model to understand handshapes, we attempt 3D hand normalization.

Our experiments conducted on the Public DGS Corpus confirmed the efficacy of these modifications for segmentation quality. By comparing to previous work in a zero-shot setting, we demonstrate that our models generalize across signed languages and domains and that including linguistically motivated cues leads to a more robust model in this context.

Finally, we envision that the proposed model has applications in real-world data collection for signed languages. Furthermore, a similar segmentation approach could be leveraged in various other fields such as co-speech gesture recognition (Moryossef, 2023) and action segmentation (Tang et al., 2019).

# Limitations

## Pose Estimation

In this work, we employ the MediaPipe Holistic pose estimation system (Grishchenko and Bazarevsky, 2020). There is a possibility that this system exhibits bias towards certain protected classes (such as gender or race), underperforming in instances with specific skin tones or lower video quality. Thus, we cannot attest to how our system would perform under real-world conditions, given that the videos utilized in our research are generated in a controlled studio environment, primarily featuring white participants.

## Encoding of Long Sequences

In this study, we encode sequences of frames that are significantly longer than the typical 512 frames often seen in models employing Transformers (Vaswani et al., 2017). Numerous techniques, ranging from basic temporal pooling/downsampling to more advanced methods such as a video/pose encoder that aggregates local frames into higher-level 'tokens' (Renz et al., 2021a), graph convolutional networks (Bull et al., 2020b), and self-supervised representations (Baevski et al., 2020), can alleviate length constraints, facilitate the use of Transformers, and potentially improve the outcomes. Moreover, a hierarchical method like the Swin Transformer (Liu et al., 2021) could be applicable.

## Limitations of Autoregressive LSTMs

In this paper, we replicated the autoregressive LSTM implementation originally proposed by Jiang et al. (2023). Our experiments revealed that this implementation exhibits significant slowness, which prevented us from performing further experimentation. In contrast, other LSTM implementations employed in this project have undergone extensive optimization (Appleyard, 2016), including techniques like combining general matrix multiplication operations (GEMMs), parallelizing independent operations, fusing kernels, rearranging matrices, and implementing various optimizations for models with multiple layers (which are not necessarily applicable here).

A comparison of CPU-based performance demonstrates that our implementation is x6.4 times slower. Theoretically, the number of operations performed by the autoregressive LSTM is equivalent to that of a regular LSTM. However, while the normal LSTM benefits from concurrency based on the number of layers, we do not have that luxury. The optimization of recurrent neural networks (RNNs) (Que et al., 2020, 2021, 2022) remains an ongoing area of research. If proven effective in other domains, we strongly advocate for efforts to optimize the performance of this type of network.

## Interference Between Sign and Phrase Models

In our model, we share the encoder for both the sign and phrase segmentation models, with a shallow linear layer for the BIO tag prediction associated with each task. It remains uncertain whether these two tasks interfere with or enhance each other. An

ablation study (not presented in this work) involving separate modeling is necessary to obtain greater insight into this matter.

## Noisy Training Objective

Although the annotations utilized in this study are of expert level, the determination of precise sign (Hanke et al., 2012) and phrase boundaries remains a challenging task, even for experts. Training the model on these annotated boundaries might introduce excessive noise. A similar issue was observed in classification-based pose estimation (Cao et al., 2019). The task of annotating the exact anatomical centers of joints proves to be nearly impossible, leading to a high degree of noise when predicting joint position as a 1-hot classification task. The solution proposed in this previous work was to distribute a Gaussian around the annotated location of each joint. This approach allows the joint's center to overlap with some probability mass, thereby reducing the noise for the model. A similar solution could be applied in our context. Instead of predicting a strict 0 or 1 class probability, we could distribute a Gaussian around the boundary.

## Naive Segment Decoding

We recognize that the frame-level greedy decoding strategy implemented in our study may not be optimal. Previous research in audio segmentation (Venkatesh et al., 2022) employed a You Only Look Once (YOLO; Redmon et al. (2015)) decoding scheme to predict segment boundaries and classes. We propose using a similar prediction atop a given representation, such as the LSTM output or classification logits of an already trained network. Differing from traditional object detection tasks, this process is simplified due to the absence of a $Y$ axis and non-overlapping segments. In this scenario, the network predicts the segment boundaries using regression, thereby avoiding the class imbalance issue of the BIO tagging. We anticipate this to yield more accurate sign language segmentation.

## Lack of Transcription

Speech segmentation is a close task to our sign language segmentation task on videos. In addition to relying on prosodic cues from audio, the former could benefit from automatic speech transcription systems, either in terms of surrogating the task to text-level segmentation and punctuation (Cho et al., 2015), or gaining additional training data from automatic speech recognition / spoken language translation (Tsiamas et al., 2022).

However, for signed languages, there is neither a standardized and widely used written form nor a reliable transcription procedure into some potential writing systems like SignWriting (Sutton, 1990), HamNoSys (Prillwitz and Zienert, 1990), and glosses (Johnston, 2008). Transcription/recognition and segmentation tasks need to be solved simultaneously, so we envision that a multitask setting helps. Sign spotting, the localization of a specific sign in continuous signing, is a simplification of the segmentation and recognition problem in a closed-vocabulary setting (Wong et al., 2022; Varol et al., 2022). It can be used to find candidate boundaries for some signs, but not all.

## Acknowledgements

This work was funded by the EU Horizon 2020 project EASIER (grant agreement no. 101016982), the Swiss Innovation Agency (Innosuisse) flagship IICT (PFFS-21-47) and the EU Horizon 2020 project iEXTRACT (grant agreement no. 802774). We also thank Rico Sennrich and Chantal Amrhein for their suggestions.

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

# A  Extended Experimental Results

We conducted some preliminary experiments (starting with *P0*) on training a sign language segmentation model to gain insights into hyperparameters and feature choices. The results are shown in Table 3[12]. We found in *P1.3.2* the optimal hyperparameters and repeated them with different feature choices.

| Experiment | | | Sign | | | Phrase | | |
|---|---|---|---|---|---|---|---|---|
| | | | F1 | IoU | % | F1 | IoU | % |
| P0 | Moryossef et al. (2020) | test | — | 0.4 | 1.45 | — | 0.65 | 0.82 |
| | | dev | — | 0.35 | 1.36 | — | 0.6 | 0.77 |
| P0.1 | P0 + Holistic 25fps | test | — | 0.39 | 0.86 | — | 0.64 | 0.5 |
| | | dev | — | 0.32 | 0.81 | — | 0.58 | 0.52 |
| P1 | P1 baseline | test | 0.55 | 0.49 | 0.83 | 0.6 | 0.67 | 2.63 |
| | | dev | 0.56 | 0.43 | 0.75 | 0.58 | 0.62 | 2.61 |
| P1.1 | P1 - encoder_bidirectional | test | 0.48 | 0.45 | 0.68 | 0.5 | 0.64 | 2.68 |
| | | dev | 0.46 | 0.41 | 0.64 | 0.51 | 0.61 | 2.56 |
| P1.2.1 | P1 + hidden_dim=512 | test | 0.47 | 0.42 | 0.44 | 0.52 | 0.63 | 1.7 |
| | | dev | 0.46 | 0.4 | 0.43 | 0.52 | 0.61 | 1.69 |
| P1.2.2 | P1 + hidden_dim=1024 | test | 0.48 | 0.45 | 0.42 | 0.58 | 0.65 | 1.53 |
| | | dev | 0.46 | 0.41 | 0.36 | 0.53 | 0.61 | 1.49 |
| P1.3.1 | P1 + encoder_depth=2 | test | 0.55 | 0.48 | 0.76 | 0.58 | 0.67 | 2.56 |
| | | dev | 0.56 | 0.43 | 0.69 | 0.58 | 0.62 | 2.52 |
| P1.3.2 | P1 + encoder_depth=4 | test | 0.63 | 0.51 | 0.91 | 0.66 | 0.67 | 1.41 |
| | | dev | 0.61 | 0.47 | 0.84 | 0.64 | 0.6 | 1.39 |
| P1.4.1 | P1 + hidden_dim=128 + encoder_depth=2 | test | 0.58 | 0.48 | 0.8 | 0.6 | 0.67 | 2.0 |
| | | dev | 0.55 | 0.43 | 0.75 | 0.54 | 0.62 | 2.03 |
| P1.4.2 | P1 + hidden_dim=128 + encoder_depth=4 | test | 0.62 | 0.51 | 0.91 | 0.64 | 0.68 | 2.43 |
| | | dev | 0.6 | 0.47 | 0.83 | 0.6 | 0.62 | 2.57 |
| P1.4.3 | P1 + hidden_dim=128 + encoder_depth=8 | test | 0.59 | 0.52 | 0.91 | 0.63 | 0.68 | 3.04 |
| | | dev | 0.6 | 0.47 | 0.84 | 0.6 | 0.62 | 3.02 |
| P1.5.1 | P1 + hidden_dim=64 + encoder_depth=4 | test | 0.57 | 0.5 | 0.8 | 0.6 | 0.68 | 2.41 |
| | | dev | 0.58 | 0.45 | 0.75 | 0.59 | 0.62 | 2.39 |
| P1.5.2 | P1 + hidden_dim=64 + encoder_depth=8 | test | 0.62 | 0.51 | 0.85 | 0.64 | 0.68 | 2.53 |
| | | dev | 0.6 | 0.46 | 0.79 | 0.6 | 0.62 | 2.53 |
| P2 | P1 + optical_flow | test | 0.58 | 0.5 | 0.95 | 0.63 | 0.68 | 3.17 |
| | | dev | 0.59 | 0.45 | 0.84 | 0.59 | 0.61 | 3.08 |
| P2.1 | P1.3.2 + optical_flow | test | 0.63 | 0.51 | 0.92 | 0.66 | 0.67 | 1.51 |
| | | dev | 0.62 | 0.46 | 0.81 | 0.62 | 0.6 | 1.53 |
| P3 | P1 + hand_normalization | test | 0.55 | 0.48 | 0.77 | 0.58 | 0.67 | 2.79 |
| | | dev | 0.55 | 0.42 | 0.71 | 0.57 | 0.62 | 2.73 |
| P3.1 | P1.3.2 + hand_normalization | test | 0.63 | 0.51 | 0.91 | 0.66 | 0.66 | 1.43 |
| | | dev | 0.61 | 0.46 | 0.82 | 0.64 | 0.61 | 1.46 |
| P4 | P2.1 + P3.1 | test | 0.56 | 0.51 | 0.92 | 0.61 | 0.66 | 1.45 |
| | | dev | 0.61 | 0.46 | 0.81 | 0.63 | 0.6 | 1.41 |
| P4.1 | P4 + encoder_depth=8 | test | 0.6 | 0.51 | 0.95 | 0.62 | 0.67 | 1.08 |
| | | dev | 0.61 | 0.47 | 0.86 | 0.62 | 0.6 | 1.12 |
| P5 | P1.3.2 + reduced_face | test | 0.63 | 0.51 | 0.94 | 0.64 | 0.66 | 1.16 |
| | | dev | 0.61 | 0.47 | 0.86 | 0.64 | 0.58 | 1.14 |
| P5.1 | P1.3.2 + full_face | test | 0.54 | 0.49 | 0.8 | 0.6 | 0.68 | 2.29 |
| | | dev | 0.57 | 0.45 | 0.7 | 0.59 | 0.62 | 2.29 |

Table 3: Results of the preliminary experiments.

---

[12]Note that due to an implementation issue on edge cases (which we fixed later), the *IoU* and % values in Table 3 are lower than the ones in Table 1 and Table 4 thus not comparable across tables. The comparison inside of Table 3 between different experiments remains meaningful. In addition, the results in Table 3 are based on only one run instead of three random runs.

We selected some promising models from our preliminary experiments and reran them three times using different random seeds to make the final conclusion reliable and robust. Table 4 includes the standard deviation and the validation results (where we performed the model selection) for readers to scrutinize.

| Experiment | | | Sign | | | Phrase | | | Efficiency | |
|---|---|---|---|---|---|---|---|---|---|---|
| | | | F1 | IoU | % | F1 | IoU | % | #Params | Time |
| E0 | Moryossef et al. (2020) | test | — | $0.46 \pm 0.03$ | $1.09 \pm 0.41$ | — | $0.70 \pm 0.01$ | $1.00 \pm 0.06$ | 102K | 0:50:17 |
| | | dev | — | $0.42 \pm 0.05$ | $1.21 \pm 0.59$ | — | $0.61 \pm 0.06$ | $2.47 \pm 0.85$ | 102K | 0:50:17 |
| E1 | Baseline | test | $0.56 \pm 0.03$ | $0.66 \pm 0.01$ | $0.91 \pm 0.05$ | $0.59 \pm 0.02$ | $0.80 \pm 0.03$ | $2.50 \pm 0.13$ | 454K | 1:01:50 |
| | | dev | $0.55 \pm 0.01$ | $0.59 \pm 0.00$ | $1.12 \pm 0.11$ | $0.56 \pm 0.02$ | $0.75 \pm 0.05$ | $2.94 \pm 0.08$ | 454K | 1:01:50 |
| E2 | E1 + Face | test | $0.53 \pm 0.05$ | $0.58 \pm 0.07$ | $0.64 \pm 0.30$ | $0.57 \pm 0.02$ | $0.76 \pm 0.03$ | $1.87 \pm 0.83$ | 552K | 1:50:31 |
| | | dev | $0.50 \pm 0.07$ | $0.53 \pm 0.11$ | $0.90 \pm 0.19$ | $0.53 \pm 0.05$ | $0.71 \pm 0.07$ | $2.43 \pm 1.02$ | 552K | 1:50:31 |
| E3 | E1 + Optical Flow | test | $0.58 \pm 0.01$ | $0.62 \pm 0.00$ | $1.12 \pm 0.05$ | $0.60 \pm 0.03$ | $0.82 \pm 0.03$ | $3.19 \pm 0.11$ | 473K | 1:20:17 |
| | | dev | $0.58 \pm 0.00$ | $0.62 \pm 0.00$ | $1.50 \pm 0.19$ | $0.59 \pm 0.01$ | $0.79 \pm 0.00$ | $3.94 \pm 0.14$ | 473K | 1:20:17 |
| E4 | E3 + Hand Norm | test | $0.56 \pm 0.02$ | $0.61 \pm 0.00$ | $1.07 \pm 0.05$ | $0.60 \pm 0.00$ | $0.80 \pm 0.00$ | $3.24 \pm 0.17$ | 516K | 1:30:59 |
| | | dev | $0.57 \pm 0.01$ | $0.61 \pm 0.01$ | $1.50 \pm 0.07$ | $0.58 \pm 0.00$ | $0.79 \pm 0.00$ | $4.04 \pm 0.31$ | 516K | 1:30:59 |
| E1s | E1 + Depth=4 | test | $0.63 \pm 0.01$ | $0.69 \pm 0.00$ | $1.11 \pm 0.01$ | $0.65 \pm 0.02$ | $0.82 \pm 0.04$ | $1.63 \pm 0.10$ | 1.6M | 4:08:48 |
| | | dev | $0.61 \pm 0.00$ | $0.63 \pm 0.00$ | $1.27 \pm 0.01$ | $0.63 \pm 0.01$ | $0.77 \pm 0.01$ | $2.17 \pm 0.18$ | 1.6M | 4:08:48 |
| E2s | E2 + Depth=4 | test | $0.62 \pm 0.02$ | $0.69 \pm 0.00$ | $1.07 \pm 0.03$ | $0.63 \pm 0.01$ | $0.84 \pm 0.03$ | $2.68 \pm 0.53$ | 1.7M | 3:14:03 |
| | | dev | $0.60 \pm 0.01$ | $0.63 \pm 0.01$ | $1.20 \pm 0.12$ | $0.59 \pm 0.02$ | $0.76 \pm 0.05$ | $3.30 \pm 0.62$ | 1.7M | 3:14:03 |
| E3s | E3 + Depth=4 | test | $0.60 \pm 0.01$ | $0.63 \pm 0.00$ | $1.13 \pm 0.01$ | $0.64 \pm 0.03$ | $0.80 \pm 0.03$ | $1.53 \pm 0.18$ | 1.7M | 4:08:30 |
| | | dev | $0.62 \pm 0.00$ | $0.63 \pm 0.00$ | $1.63 \pm 0.05$ | $0.63 \pm 0.00$ | $0.76 \pm 0.00$ | $2.14 \pm 0.09$ | 1.7M | 4:08:30 |
| E4s | E4 + Depth=4 | test | $0.59 \pm 0.00$ | $0.63 \pm 0.00$ | $1.13 \pm 0.03$ | $0.62 \pm 0.00$ | $0.79 \pm 0.00$ | $1.43 \pm 0.10$ | 1.7M | 4:35:29 |
| | | dev | $0.61 \pm 0.00$ | $0.63 \pm 0.00$ | $1.56 \pm 0.04$ | $0.63 \pm 0.00$ | $0.77 \pm 0.01$ | $1.89 \pm 0.07$ | 1.7M | 4:35:29 |
| E4ba | E4s + Autoregressive | test | $0.45 \pm 0.03$ | $0.47 \pm 0.05$ | $0.88 \pm 0.08$ | $0.52 \pm 0.02$ | $0.63 \pm 0.10$ | $2.72 \pm 1.33$ | 1.3M | 2 days, 21:28:42 |
| | | dev | $0.40 \pm 0.01$ | $0.40 \pm 0.01$ | $2.02 \pm 0.73$ | $0.47 \pm 0.00$ | $0.57 \pm 0.04$ | $4.26 \pm 1.26$ | 1.3M | 2 days, 21:28:42 |

Table 4: Mean evaluation metrics for our main experiments. A complete version of Table 1.

## B  Greedy Decoding Algorithm

We provide our exact decoding algorithm in Algorithm 1. We opt to employ adjustable thresholds rather than *argmax* prediction, as our empirical findings demonstrate superior performance with this approach (§5.2).

---

**Algorithm 1** Probabilities to Segments Conversion.

---

**Require:** $probs$, a list of probabilities from 0 to 100
1:  $threshold_b \leftarrow 50.0$
2:  $threshold_o \leftarrow 50.0$
3:
4:  $start \leftarrow None$
5:  $did\_pass\_start \leftarrow False$
6:
7:  **for** $i = 0$ **to** $len(probs)$ **do**
8:      $b, i, o \leftarrow probs[i]$
9:
10:      **if** $start = None$ **then**
11:          **if** $b > threshold_b$ **then**
12:              $start \leftarrow i$
13:          **end if**
14:      **else**
15:          **if** $did\_pass\_start$ **then**
16:              **if** $b > threshold_b$ **or** $o > threshold_o$ **then**
17:                  **yield** $(start, i - 1))$
18:                  $start \leftarrow None$
19:                  $did\_pass\_start \leftarrow False$
20:              **end if**
21:          **else**
22:              **if** $b < threshold_b$ **then**
23:                  $did\_pass\_start \leftarrow True$
24:              **end if**
25:          **end if**
26:      **end if**
27:  **end for**
28:
29:  **if** $start \neq None$ **then**
30:      **yield** $(start, len(probs)))$
31:  **end if**

---

## C Pose Based Hand Shape Analysis

### C.1 Introduction to Hand Shapes in Sign Language

The most prominent feature of signed languages is their use of the hands. In fact, the hands play an integral role in the phonetics of signs, and a slight variation in hand shape can convey differences in meaning (Stokoe Jr, 1960). In sign languages such as American Sign Language (ASL) and British Sign Language (BSL), different hand shapes contribute to the vocabulary of the language, similar to how different sounds contribute to the vocabulary of spoken languages. ASL is estimated to use between 30 to 80 hand shapes[13], while BSL is limited to approximately 40 hand shapes[14]. SignWriting (Sutton, 1990), a system of notation used for sign languages, specifies a superset of 261 distinct hand shapes (Frost and Sutton, 2022). Each sign language uses a subset of these hand shapes.

Despite the fundamental role of hand shapes in sign languages, accurately recognizing and classifying them is a challenging task. In this section, we explore rule-based hand shape analysis in sign languages using 3D hand normalization. By performing 3D hand normalization, we can transform any given hand shape to a fixed orientation, making it easier for a model to extract the hand shape, and hence improving the recognition and classification of hand shapes in sign languages.

### C.2 Characteristics of the Human Hand

The human hand consists of 27 bones and can be divided into three main sections: the wrist (carpals), the palm (metacarpals), and the fingers (phalanges). Each finger consists of three bones, except for the thumb, which has two. The bones are connected by joints, which allow for the complex movements and shapes that the hand can form.

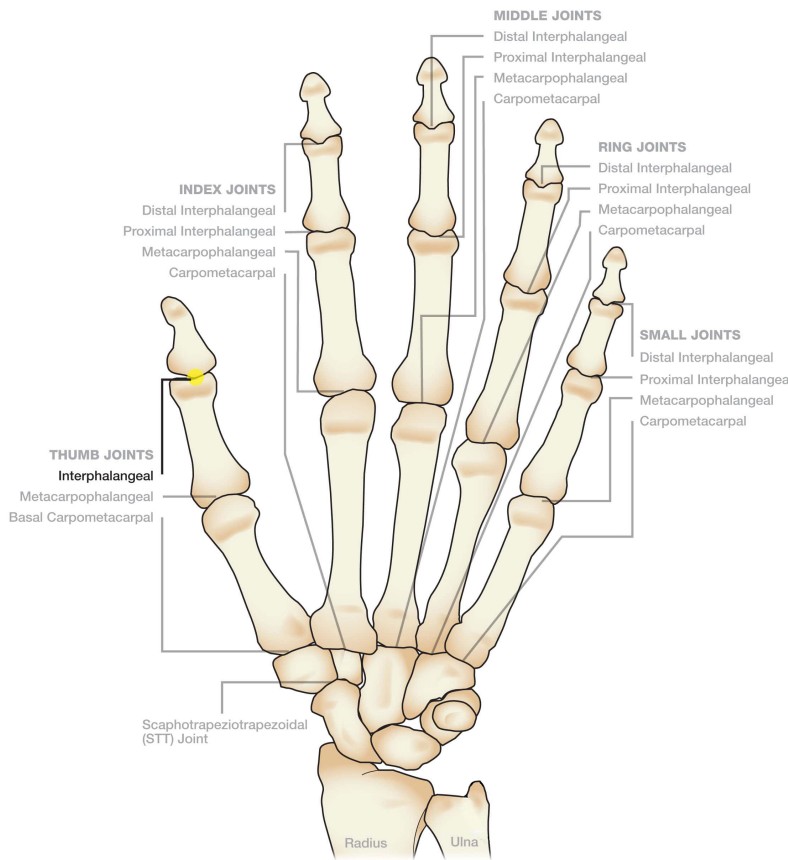

Figure 7: Anatomy of a human hand. ©American Society for Surgery of the Hand

---

[13]https://aslfont.github.io/Symbol-Font-For-ASL/asl/handshapes.html
[14]https://bsl.surrey.ac.uk/principles/i-hand-shapes

Understanding the different characteristics of hands and their implications in signed languages is crucial for the extraction and classification of hand shapes. These characteristics are based on the SignWriting definitions of the five major axes of hand variation: handedness, plane, rotation, view, and shape.

**Handedness** is the distinction between the right and left hands. Signed languages make a distinction between the dominant hand and the non-dominant hand. For right-handed individuals, the right hand is considered dominant, and vice-versa. The dominant hand is used for fingerspelling and all one-handed signs, while the non-dominant hand is used for support and two-handed signs. Using 3D pose estimation, the handedness analysis is trivial, as the pose estimation platform predicts which hand is which.

**Plane** refers to whether the hand is parallel to the wall or the floor. The variation in the plane can, but does not have to, create a distinction between two signs. For example, in ASL the signs for "date" and "dessert" exhibit the same hand shape, view, rotation, and movement, but differ by plane. The plane of a hand can be estimated by comparing the positions of the wrist and middle finger metacarpal bone ($M\_MCP$).

---

**Algorithm 2** Hand Plane Estimation

---

1: $y \leftarrow |M\_MCP.y - WRIST.y| \times 1.5$ // add bias to y
2: $z \leftarrow |M\_MCP.z - WRIST.z|$
3: **return** $y > z$ ? 'wall' : 'floor'

---

**Rotation** refers to the angle of the hand in relation to the body. SignWriting groups the hand rotation into eight equal categories, each spanning 45 degrees. The rotation of a hand can be calculated by finding the angle of the line created by the wrist and the middle finger metacarpal bone.

**View** refers to the side of the hand as observed by the signer, and is grouped into four categories: front, back, sideways, and other-sideways. The view of a hand can be estimated by analyzing the normal of the plane created by the palm of the hand (between the wrist, index finger metacarpal bone, and pinky metacarpal bone).

---

**Algorithm 3** Hand View Estimation

---

1: normal $\leftarrow$ math.normal(WRIST, I\_MCP, P\_MCP)
2: plane $\leftarrow$ get_plane(WRIST, M\_MCP)
3: **if** plane = 'wall' **then**
4:      $angle \leftarrow \angle(normal.z, normal.x)$
5:      **return** $angle > 210$ ? 'front' : ($angle > 150$ ? 'sideways' : 'back')
6: **else**
7:      $angle \leftarrow \angle(normal.y, normal.x)$
8:      **return** $angle > 0$ ? 'front' : ($angle > -60$ ? 'sideways' : 'back')
9: **end if**

---

**Shape** refers to the configuration of the fingers and thumb. This characteristic of the hand is the most complex to analyze due to the vast array of possible shapes the human hand can form. The shape of a hand is determined by the state of each finger and thumb, specifically whether they are straight, curved, or bent, and their position relative to each other. Shape analysis can be accomplished by examining the bend and rotation of each finger joint. More advanced models may also take into consideration the spread between the fingers and other nuanced characteristics. 3D pose estimation can be used to extract these features for a machine learning model, which can then classify the hand shape.

### C.3 3D Hand Normalization

3D hand normalization is an attempt at standardizing the orientation and position of the hand, thereby enabling models to effectively classify various hand shapes. The normalization process involves several

key steps, as illustrated below:

1. **Pose Estimation** Initially, the 3D pose of the hand is estimated from the hand image crop (Figure 8).

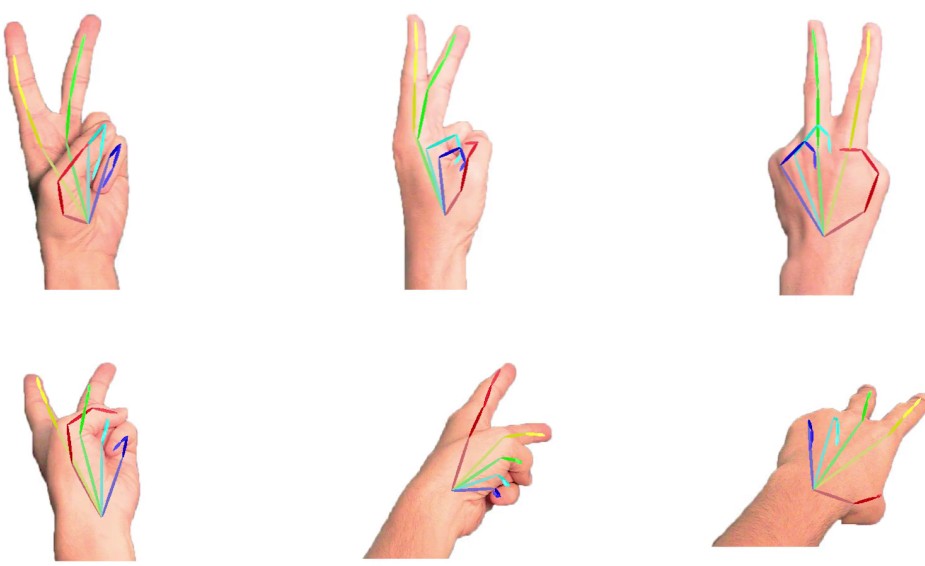

Figure 8: Pictures of six hands all performing the same hand shape (v-shape) taken from six different orientations. Mediapipe fails at estimating the pose of the bottom-middle image.

2. **3D Rotation** The pose is then rotated in 3D space such that the normal of the back of the hand aligns with the $Z$-axis. As a result, the palm plane now resides within the $XY$ plane (Figure 9).

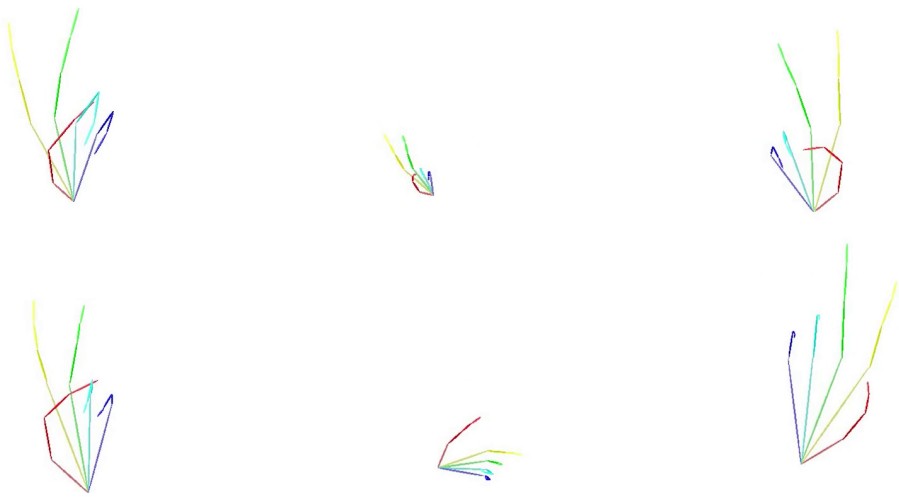

Figure 9: Hand poses after 3D rotation. The scale difference between the hands demonstrates a limitation of the 3D pose estimation system used.

3. **2D Orientation** Subsequently, the pose is rotated in 2D such that the metacarpal bone of the middle finger aligns with the $Y$-axis (Figure 10).

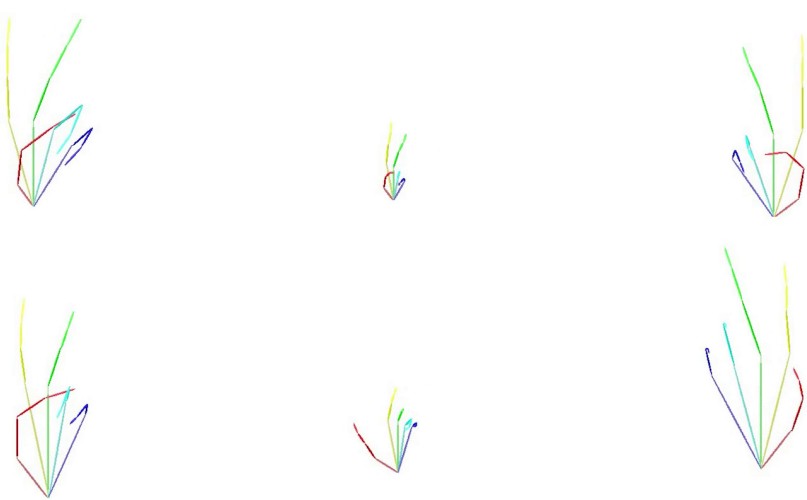

Figure 10: Hand poses after being rotated.

4. **Scale** The hand is scaled such that the metacarpal bone of the middle finger attains a constant length (which we typically set to 200, Figure 11).

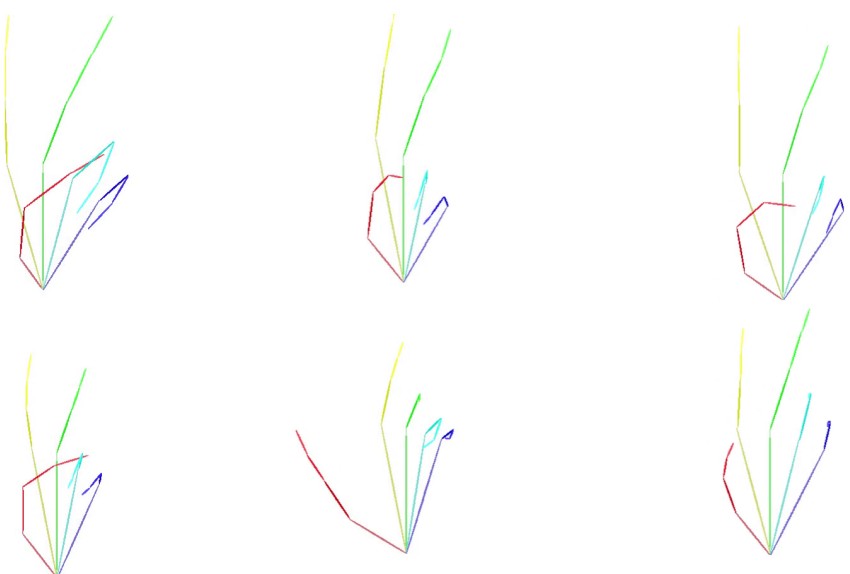

Figure 11: Hand poses after being scaled.

5. **Translation** Lastly, the wrist joint is translated to the origin of the coordinate system $(0, 0, 0)$. Figure 12 demonstrates how when overlayed, we can see all hands producing the same shape, except for one outlier.

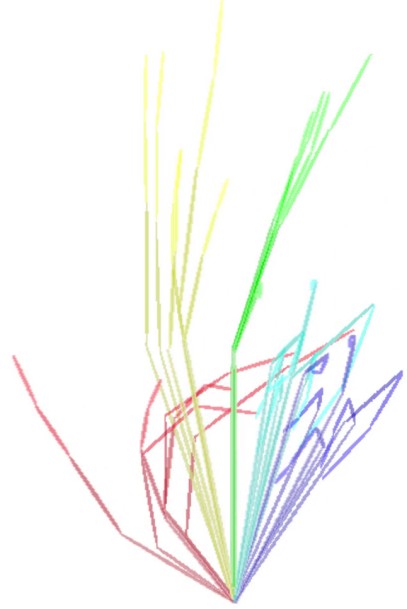

Figure 12: Normalized hand poses overlayed after being translated to the same position. The positions of the wrist and the metacarpal bone of the middle finger are fixed.

By conducting these normalization steps, a hand model can be standardized, reducing the complexity of subsequent steps such as feature extraction and hand shape classification. This standardization simplifies the recognition process and can contribute to improving the overall accuracy of the system.

**C.4 3D Hand Pose Evaluation**

In order to assess the performance of our 3D hand pose estimation and normalization, we introduce two metrics that gauge the consistency of the pose estimation across orientations and crops.

Our dataset is extracted from the SignWriting Hand Symbols Manual (Frost and Sutton, 2022), and includes images of 261 different hand shapes, from 6 different angles. All images are of the same hand, of an adult white man.

**Multi Angle Consistency Error (MACE)** evaluates the consistency of the pose estimation system across the different orientations. We perform 3D hand normalization, and overlay the hands. The MACE score is the average standard deviation of all pose landmarks, between all views. A high MACE score indicates a problem in the pose estimation system's ability to maintain consistency across different orientations. This could adversely affect the model's performance when analyzing hand shapes in sign languages, as signs can significantly vary with hand rotation.

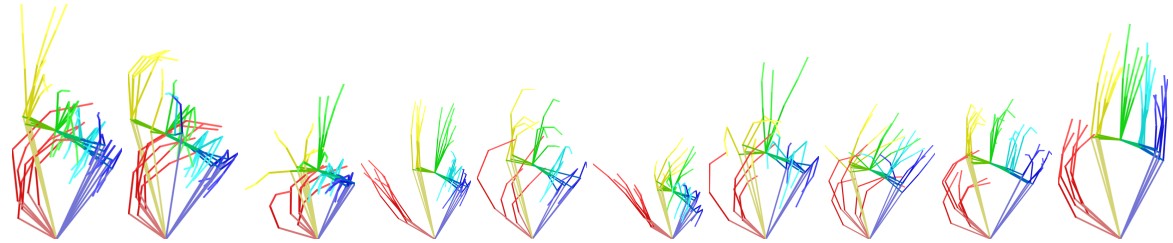

Figure 13: Visualizations of 10 hand shapes, each with 6 orientations 3D normalized and overlayed.

Figure 13 shows that our 3D normalization does work to some extent using Mediapipe. We can identify differences across hand shapes, but still note high variance within each hand shape.

**Crop Consistency Error (CCE)**   gauges the pose estimation system's consistency across different crop sizes. We do not perform 3D normalization, but still overlay all the estimated hands, shifting the wrist point of each estimated hand to the origin $(0, 0, 0)$. The CCE score is the calculated average standard deviation of all pose landmarks across crops. A high CCE score indicates that the pose estimation system is sensitive to the size of the input crop, which is a significant drawback as the system should be invariant to the size of the input image.

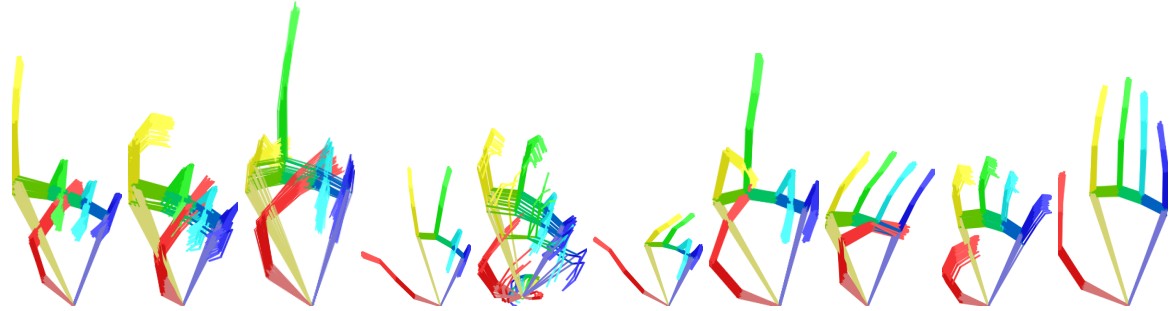

Figure 14: Visualizations of 10 hand shapes, each with 48 crops overlayed.

Figure 14 shows that for some poses, Mediapipe is very resilient to crop size differences (e.g. the first and last hand shapes). However, it is concerning that for some hand shapes, it exhibits very high variance, and possibly even wrong predictions.

## C.5  Conclusion

Our normalization process appears to work reasonably well when applied to different views within the same crop size. It succeeds in simplifying the hand shape, which in turn, can aid in improving the accuracy of hand shape classification systems.

However, it is crucial to note that while this method may seem to perform well on a static image, its consistency and reliability in a dynamic context, such as a video, may be quite different. In a video, the crop size can change between frames, introducing additional complexity and variance. This dynamic nature coupled with the inherently noisy nature of the estimation process can pose challenges for a model that aims to consistently estimate hand shapes.

In light of these findings, it is clear that there is a need for the developers of 3D pose estimation systems to consider these evaluation methods and strive to make their systems more robust to changes in hand crops. The Multi Angle Consistency Error (MACE) and the Crop Consistency Error (CCE) can be valuable tools in this regard.

MACE could potentially be incorporated as a loss function for 3D pose estimation, thereby driving the model to maintain consistency across different orientations. Alternatively, MACE could be used as an indicator to identify hand shapes that require more training data. It is apparent from our study that the performance varies greatly across hand shapes and orientations, and this approach could help in prioritizing the allocation of training resources.

Ultimately, the goal of improving 3D hand pose estimation is to enhance the ability to encode signed languages accurately. The insights gathered from this study can guide future research and development efforts in this direction, paving the way for more robust and reliable sign language technology.

The benchmark, metrics, and visualizations are available at https://github.com/sign-language-processing/3d-hands-benchmark/.