# OpenReview forum: "Linguistically Motivated Sign Language Segmentation"
_EMNLP/2023/Conference — EMNLP 2023 Findings_

### Official Review · Reviewer_7cdW · 2023-07-25

**Soundness:** 2

**Excitement:**

2: Mediocre: This paper makes marginal contributions (vs non-contemporaneous work), so I would rather not see it in the conference.

**Paper Topic And Main Contributions:**

The paper proposes a BIO frame tagging method for sequences and single signs within those sequences. Authors argue that it is beneficial for segmentation tasks.

The paper proposes a 3D normalization method for handshapes using MediaPipe's hand model.

**Questions For The Authors:**

A.Why not use additional publicly available corpora to increase training data and generalization? Aside from DGS, there are other European corpora (Swedish, Greek, etc.)
B.The seq2seq and transformer models use attention and/or positional encoding, making the BIO tagging unnecessary. How would you defend your work against this statement?

**Reasons To Accept:**

The proposed model can be used in the outside domain for further data collection in the wild.

**Reasons To Reject:**

Experimental results do not show an improvement over the suggested baseline E1, arguing against one of the contributions of the paper (Hand Norm)
Reported Stds in Table 4 indicate that reported performance differences might be insignificant.
The analysis of the results needs to be more comprehensive. Why does adding facial expression data lowers the performance, and why does not the proposed hand norm benefit it as well?
The paper has an extra six pages long second paper in Appendix section C, with no obvious justification. Author could just mention that they perform a linear transformation of 3D coordinates for normalization, that would be sufficient.

**Reproducibility:**

4: Could mostly reproduce the results, but there may be some variation because of sample variance or minor variations in their interpretation of the protocol or method.

**Reviewer Confidence:**

2: Willing to defend my evaluation, but it is fairly likely that I missed some details, didn't understand some central points, or can't be sure about the novelty of the work.

**Typos Grammar Style And Presentation Improvements:**

In the context of SL, more suitable terms are:
word -> sign, gloss, or annotated unit/token
phrase -> utterance  (line 216)

Reproducibility, Ethics Review, Anonymity Requirement and Overall Recommendation
https://github.com/anon/anonymous
https://github.com/anon/anonymous2

Both links did not work - returned 404; is that a placeholder?

---

> ### Author Rebuttal · Authors · 2023-08-27
>
> ### Rebuttal
>
> #### Limited Improvement Over Baseline and Inconsistent Standard Deviations
> **Reviewer’s Comment:**
> The experimental results do not show significant improvements over the baseline and questions the standard deviations in Table 4.
>
> **Response:**
> The results given by E2s, E3s, and E4s indeed are not an improvement over E1s in the in-domain setting.
> The added features, like optical flow, do improve the performance in shallower models. In deeper models, they offer more than just trivial in-domain improvements; they crucially enhance the model's adaptability to new contexts and languages (Table 2).
>
> ---
>
> #### Analysis of Facial Expression Data and Hand Normalization
> **Reviewer’s Comment:**
> Why the inclusion of facial expression data and hand normalization did not lead to performance benefits.
>
> **Response:**
> For the face points, although the face is an essential component to understanding sign language experessions, and does play some role of it in sign and phrase level segmentation, we believe that the 128 face points are too dense for the model to learn useful information compared to the 75 body points, and may on the opppsite, confuse the model. We will include more speculations on this line in our paper.
> While the use of 3D hand normalization is well-justified in the paper, it does not help the model, due to poor depth estimation quality, as further corroborated by recent research [De Coster et al. 2023](https://arxiv.org/pdf/2306.17558.pdf). Therefore, we consider it a negative result, showing the deficiencies in the 3D pose estimation system. This analysis will be further clarified in the camera ready.
>
> ---
>
> #### Length of Appendix and Justification
> **Reviewer’s Comment:**
> You comment on the extra-long appendix and its lack of obvious justification.
>
> **Response:**
> You have a very valid point. The intention was to provide comprehensive information, including quality estimation of our methodology, but we realize this may have led to unnecessary complexity. We will consider publishing the appendix separately as part of the technical documentation of the benchmark we are introducing, and focus on providing only essential information in the revised version of the manuscript.
>
> ---
>
> #### Questions for the Authors
> **Reviewer’s Questions:**
> 1. Why not use additional publicly available corpora for training?
>
> **Response:**
> Despite there being other sign language datasets publicly available, the training of sign-level segmentation requires accurate sign-level gloss annotation from continuous signing, and the training of phrase-level segmentation requires perfect alignment between sign language video and spoken language sentences, which is rarely the case in most datasets except the Public DGS Corpus dataset we use.
> We do highlight two previous works and the dataset each use fulfills one of the requirements above, so we also compare our models to one of them on phrase-level segmentation. In the end, the models we trained show surprisingly good zero-shot performance even without seeing additional data in a different signed language.
>
> **Reviewer’s Questions:**
> 2. How to defend the necessity of BIO tagging against seq2seq and transformer models?
>
> **Response:**
> We are not entirely sure about the question, but we will try to clarify that the use of BIO tagging is about labelling the video data with the sign/phrase start and end time, so it is irrelavant to what model archtecture we use.
> If the question is more about why we did not use a seq2seq and transformer model, the short answer is because the sequence of video frames is very long, and we have more detailed eleboration of this point in our limitation section “Encoding of Long Sequences”.
>
> ---
>
> #### Reproducibility
> We believe our work is highly reproducible. The code is available on GitHub (the links are broken due to the anonymity requirements), including all of the settings to train each model and variation presented in this paper.

---

### Official Review · Reviewer_ha7w · 2023-08-04

**Soundness:** 4

**Excitement:**

3: Ambivalent: It has merits (e.g., it reports state-of-the-art results, the idea is nice), but there are key weaknesses (e.g., it describes incremental work), and it can significantly benefit from another round of revision. However, I won't object to accepting it if my co-reviewers champion it.

**Paper Topic And Main Contributions:**

The work suggest a method for automatic segmentation of sign language. The novelty in the paper is the use BIO tagging scheme, using prosodic cues for marking phrase boundaries and using 3D hand normalization. To test their methods the authors conducted several experiments and obtained strong results.

**Questions For The Authors:**

In table 2, you describe how E1s and E4s perform on the LSF and DGS datasets. are those modes have the best results on those dataset? I would like to see the performance of other experiments just to see how well they generalize.

**Reasons To Accept:**

The experimental setup and the results are solid. The "efficiency" metric is very useful.
The idea of using 3D hand gestures is original.
Overall speaking, this paper is well-written and easy to follow.
I had some questions, but all of them were answered in the limitations section.

**Reasons To Reject:**

The idea of using BIO instead of IO is very predictable.
The study is very specific to sign language, I don't see how to use it in other tasks.
other then that, I do not see a clear reason to reject it.

**Reproducibility:**

5: Could easily reproduce the results.

**Reviewer Confidence:**

4: Quite sure. I tried to check the important points carefully. It's unlikely, though conceivable, that I missed something that should affect my ratings.

---

> ### Author Rebuttal · Authors · 2023-08-27
>
> ### Rebuttal
>
> #### Predictability of Using BIO Tagging
> **Reviewer’s Comment:**
> The use of BIO tagging instead of IO tagging is very predictable.
>
> **Response:**
> While we acknowledge that BIO tagging itself isn't new, its application to sign language segmentation addresses specific challenges unaddressed thus far by the current literature.
>
> ---
>
> #### Applicability Beyond Sign Language
> **Reviewer’s Comment:**
> The study is specific to sign language and does not see its applicability in other tasks.
>
> **Response:**
> Working on sign language should be a sufficient motivation. However, the segmentation approach could be leveraged in various other fields:
>
> 1. **Gesture Segmentation in Spoken Language**: Spoken language often incorporates gestures and facial expressions, integral to human communication, enriching the semantic layer and aiding in emotional connection. Our segmentation is directly applicable here, allowing for a better analysis of this blend of verbal and non-verbal cues.
> 2. **Action Segmentation**: In computer vision, breaking down a video into a sequence of actions is crucial for several applications. Our approach could be adapted to tackle this issue effectively. [Tang et al. 2019](https://arxiv.org/abs/1903.02874)
> 3. **Speech Segmentation**: In scenarios where segmentation is performed without the accompanying transcription step. Our methods could be applied in these contexts.
>
> ---
>
> #### Questions for the Authors
> **Reviewer’s Question:**
> You are interested in knowing how different experiments performed on the LSF and DGS datasets, questioning if E1s and E4s have the best results on those datasets.
>
> **Response:**
> Since E1s and E4s are the most different (E4s includes all our engineered features), we find it interesting to apply them on another unseen dataset and compare them.
> We posit our results in Table 2 specifically as a zero-shot, out-of-domain generalization. While the datasets are annotated differently, we can still see the improvement from our feature engineering, most importantly in F1 score, which leads us to believe the features we added are indeed important for out-of-domain generalization.
> However, we agree that including results trained on the MEDIAPI-SKEL dataset could offer additional insights. We will incorporate this into the revised paper.

---

### Official Review · Reviewer_i9D9 · 2023-08-05

**Soundness:** 2

**Excitement:**

2: Mediocre: This paper makes marginal contributions (vs non-contemporaneous work), so I would rather not see it in the conference.

**Paper Topic And Main Contributions:**

The paper proposes a new scheme of sign language segmentation, which include two aspects: replacing the predominant IO tagging scheme with BIO tagging to account for continuous signing and segmenting phrases as well as segmenting signs. The authors also propose a method using optical flow and 3D hand normalization. Experimental results show the proposed method outperforms a baseline system.

**Reasons To Accept:**

The paper proposes a new scheme of sign language segmentation, which include two aspects: replacing the predominant IO tagging scheme with BIO tagging to account for continuous signing and segmenting phrases as well as segmenting signs. By using the new scheme, better segmentation performance may be achieved.

**Reasons To Reject:**

1. Novelty seems insufficient. Although replacing the IO tagging scheme with BIO tagging may benefit sign language segmentation, the idea is not completely novel (as the authors mentioned, it is proposed in 1995 for similar NLP tasks.
2. Experimental results seems not convincing enough. Only one baseline is compared to. Is that the SOTA model? Comparison to more methods or models may be needed.
3. The improvement brought by optical flow and 3D hand norm seems trivial. And by using them, the IoU and percentage of segments even drops. Is there any analysis or explanation about this?
4. Results in Table 2 cannot demonstrate the superiority of the proposed method, since the conditions are different. Is it possible to train the proposed method with the MEDIAPI-SKEL dataset?

**Reproducibility:**

5: Could easily reproduce the results.

**Reviewer Confidence:**

3: Pretty sure, but there's a chance I missed something. Although I have a good feel for this area in general, I did not carefully check the paper's details, e.g., the math, experimental design, or novelty.

---

> ### Author Rebuttal · Authors · 2023-08-27
>
> ### Rebuttal
>
>
> #### Insufficient Novelty
> **Reviewer’s Comment:**
> Replacing IO tagging with BIO tagging is not novel, given that it was proposed in 1995 for similar NLP tasks.
>
> **Response:**
>
> Our innovation is the application of BIO tagging to sign language segmentation—a context where it has not been used before. This adaptation addresses unique challenges and extends BIO tagging from text to the continuous modality of video, similar to how Vision Transformers extended Transformers to images.
>
> ---
>
> #### Experimental Results
> **Reviewer’s Comment:**
> The experimental results are not convincing enough and questions whether the baseline model is a state-of-the-art model.
>
> **Response:**
> Thank you for highlighting this concern. It is indeed the state-of-the-art and the only comparable model designed for this dataset. We will make it clearer in our paper.
> We additionally highlight two other works on other datasets in different languages that are not immediately and directly comparable, and in section 5.2 we show our trained models still achieve reasonable zero-shot performance on one of them.
>
>
> ---
>
> #### Optical Flow and 3D Hand Normalization
> **Reviewer’s Comment:**
> The improvements due to optical flow and 3D hand normalization appear to be trivial.
>
> **Response:**
> The added features, like optical flow, do improve the performance in shallower models. In deeper models, they offer more than just trivial in-domain improvements; they crucially enhance the model's adaptability to new contexts and languages (Table 2).
> While the use of 3D hand normalization is well-justified in the paper, it does not help the model, due to poor depth estimation quality, as further corroborated by recent research [De Coster et al. 2023](https://arxiv.org/pdf/2306.17558.pdf). Therefore, we consider it a negative result, showing the deficiencies in the 3D pose estimation system. This analysis will be further clarified in the camera ready.
>
>
> ---
>
> #### Results in Table 2
> **Reviewer’s Comment:**
> You question the utility of the results in Table 2 and ask whether it's possible to train the model with the MEDIAPI-SKEL dataset.
>
> **Response:**
> We posit our results in Table 2 specifically as a zero-shot, out-of-domain generalization. While the datasets are annotated differently, we can still see the improvement from our feature engineering, most importantly in ROC-AUC, which leads us to believe the features we added are indeed important for out-of-domain generalization.
> However, we agree that including results trained on the MEDIAPI-SKEL dataset could offer additional insights. We will incorporate this into the revised paper.
>
> ---
>
> #### Reproducibility
> We believe our work is highly reproducible. The code is available on GitHub (the links are broken due to the anonymity requirements), including all of the settings to train each model and variation presented in this paper.

---

### Meta-Review · Area_Chair_ehwy · 2023-09-19

**Recommendation:** 3

**Metareview:**

The paper introduces a novel approach to sign language segmentation by replacing the conventional IO tagging with BIO tagging. The authors also propose a method using optical flow and 3D hand normalization, resulting in improved segmentation quality and robustness across different signed languages and domains.
The proposed BIO tagging scheme and 3D hand normalization are discussed as potential improvements for sign language segmentation, but there are concerns about the experimental results and the significance of these contributions. The paper is well-written and easy to follow. While the use of BIO tagging is not considered highly original, the introduction of 3D hand normalization is seen as more novel. However, there are questions about the effectiveness of these contributions. While there is potential for the proposed method to be used in sign language segmentation, concerns are raised about the experimental results, the lack of a clear superiority over the baseline, and the applicability of the method to other tasks.

Pros:
- The paper is well-written and easy to follow.
- The use of 3D hand normalization is seen as a potentially original and valuable contribution.
- The paper addresses potential questions and limitations effectively.
- The proposed model may have applications in data collection for sign language in real-world settings.

Cons:
- Concerns are raised about the experimental results, including the lack of a clear superiority over the baseline and the drop in performance with the introduction of certain features, which the authors have addressed.
- The applicability of the proposed method to tasks outside of sign language segmentation is questioned but the authors have addressed.

---

### Decision · Program_Chairs · 2023-10-07

**Decision:**

Accept-Findings

**Comment:**

The paper introduces a novel approach to sign language segmentation by replacing the conventional IO tagging with BIO tagging. The authors also propose a method using optical flow and 3D hand normalization, resulting in improved segmentation quality and robustness across different signed languages and domains.
The proposed BIO tagging scheme and 3D hand normalization are discussed as potential improvements for sign language segmentation, but there are concerns about the experimental results and the significance of these contributions. The paper is well-written and easy to follow. While the use of BIO tagging is not considered highly original, the introduction of 3D hand normalization is seen as more novel. However, there are questions about the effectiveness of these contributions. While there is potential for the proposed method to be used in sign language segmentation, concerns are raised about the experimental results, the lack of a clear superiority over the baseline, and the applicability of the method to other tasks.

Pros:
- The paper is well-written and easy to follow.
- The use of 3D hand normalization is seen as a potentially original and valuable contribution.
- The paper addresses potential questions and limitations effectively.
- The proposed model may have applications in data collection for sign language in real-world settings.

Cons:
- Concerns are raised about the experimental results, including the lack of a clear superiority over the baseline and the drop in performance with the introduction of certain features, which the authors have addressed.
- The applicability of the proposed method to tasks outside of sign language segmentation is questioned but the authors have addressed.